# Different Dietary Ratios of Camelina Oil to Sandeel Oil Influence the Capacity to Synthesise and Deposit EPA and DHA in Zucker Fa/Fa Rats

**DOI:** 10.3390/nu15102344

**Published:** 2023-05-17

**Authors:** Tone-Kari Knutsdatter Østbye, Oddrun Anita Gudbrandsen, Aslaug Drotningsvik, Bente Ruyter, Gerd Marit Berge, Gjermund Vogt, Astrid Nilsson

**Affiliations:** 1Nofima AS, Norwegian Institute of Food, Fisheries and Aquaculture Research, 1433 Ås, Norway; bente.ruyter@nofima.no (B.R.); gerd.marit.berge@nofima.no (G.M.B.); astrid.nilsson@nofima.no (A.N.); 2Dietary Research Group, Centre for Nutrition, Department of Clinical Medicine, University of Bergen, 5007 Bergen, Norway; oddrun.gudbrandsen@uib.no (O.A.G.); aslaug@drotningsvik.no (A.D.); 3Vedde AS, 6030 Langevåg, Norway; 4Eurofins Food & Agro Testing Norway AS, 1538 Moss, Norway; gjermundvogt@eurofins.no

**Keywords:** ALA, DHA, EPA, Zucker rat, camelina oil, sandeel oil

## Abstract

Plant-based food provides more ALA (α-linolenic acid) and less EPA (eicosapentaenoic acid) and DHA (docosahexanoic acid) than marine food. Earlier studies indicate that cetoleic acid (22:1*n*-11) stimulates the *n*-3 pathway from ALA to EPA and DHA. The present study aimed to investigate the dietary effects of camelina oil (CA) high in ALA and sandeel oil (SA) high in cetoleic acid on the conversion of ALA to EPA and DHA. Male Zucker fa/fa rats were fed a diet of soybean oil (Ctrl) or diets of CA, SA, or a combination of CA and SA. Significantly higher levels of DPA (docosapentaenoic acid) and DHA in blood cells from the CA group compared to the Ctrl indicate an active conversion of ALA to DPA and DHA. Increasing the uptake and deposition of EPA and DHA meant that a trend towards a decrease in the liver gene expression of *Elovl5*, *Fads1*, and *Fads2* along with an increase in the dietary content of SA was observed. However, 25% of the SA could be exchanged with CA without having a significant effect on EPA, DPA, or DHA in blood cells, indicating that bioactive components in SA, such as cetoleic acid, might counteract the inhibiting effect of the high dietary content of DHA on the *n*-3 biosynthetic pathway.

## 1. Introduction

Polyunsaturated fatty acids (PUFAs) classified as *n*-3 and *n*-6, regulate a wide range of functions in the body, including blood pressure, blood clotting, and correct development and function of the brain and nervous system [1]. EPA (eicosapentaenoic acid; 20:5*n*-3) and DHA (docosahexanoic acid; 22:6*n*-3) are the predominant *n*-3 fatty acids in marine oils and their beneficial health effects are well documented [2]. The main *n*-3 fatty acid in vegetable oils is α-linolenic acid (ALA; 18:3*n*-3). Compared to EPA and DHA, the health effects of ALA have been less studied [3], but it is well known that ALA is an essential fatty acid and a precursor for EPA and DHA. The efficiency of this conversion in vivo is quite low in various species, but studies in rats indicate that DHA synthesis from ALA may be sufficient to supply the brain [4]. The DHA status in humans on vegetarian or vegan diets has been shown to be lower in men (33% versus 63%) and women (20% versus 40%) [5] than in non-vegetarian/vegan. In humans, it is shown that males and females have different conversion capacities [6]. In females, the conversion capacity of ALA to DHA was <10%, whereas the capacity in males was <3% [6,7]. Since ALA is the main *n*-3 fatty acid in plants, a shift towards a more plant-based food can result in more ALA and less EPA and DHA in the diet, and an increased knowledge of the endogenous production of EPA and DHA from ALA and the in vivo regulation of this conversion is important to the ongoing green shift with increasing numbers of flexitarians.

The long-chain *n*-3 fatty acids EPA and DHA are synthesised through multiple steps from ALA, in a pathway that is relatively well conserved between different species [8], and the same enzymes are expressed in several species including humans (reviewed by Zhang et al. [9]), rats [10], and salmon [11,12]. The pathway consists of several elongation and desaturation steps mediated by fatty acid desaturase 1 (FADS1), fatty acid desaturase 2 (FADS2), and elongases-2/5 (ELOVL2, ELOVL5) enzymes [13,14,15,16], in addition to a chain-shortening peroxisomal β-oxidation step catalysed by acyl-Coenzyme A oxidase 1. The enzymes are shared by *n*-6 and *n*-3 fatty acids producing EPA and DHA from ALA and arachidonic acid (AA; 20:4 *n*-6) and 22:4 *n*-6 from linoleic acid (LA; 18:2 *n*-6). Since LA competes with ALA for the enzymes of the biosynthetic pathway of EPA and DHA, the LA content in the diet can influence the conversion of ALA to EPA and DHA. Dietary intake of ALA has been reported to increase the EPA content in plasma phospholipid, but not the DHA content in several rat and human studies [4,17], suggesting rate-limiting steps between EPA and DHA. The efficiency of the conversion of ALA to EPA and DHA in vivo is quite low in many species but can be improved by distinct factors. Diet composition, life stage, genotype, and growth influence the capacity for EPA and DHA synthesis in different species [18,19,20,21,22,23,24,25,26,27,28]. Dietary fatty acids may exert a major influence on the efficiency of the pathway. High dietary levels of DHA can inhibit, by feedback inhibition, the *n*-3 biosynthetic pathway [27,29,30,31], whereas a lack of DHA in the diet stimulates the pathway [32,33,34].

Several species of the Brassicaceae family are grown for dietary seed oil production. These oils are characterised by variable but generally large amounts of the *n*-6 and *n*-3 fatty acids LA and ALA as well as the monounsaturated fatty acids (MUFAs) oleic acid (18:1 *n*-9) and eicosenoic acid (20:1 *n*-9), which make up 95–98% of the total lipids of high quality, viable seeds [35]. Camelina (*Camelina sativa*) is an underexploited member of the Brassicaceae family commonly known as false flax and gold of pleasure [36]. Camelina oil (CA) is an *n*-3-rich vegetable oil containing up to 40% (*w*/*w*) ALA. Archaeological excavations in Europa and Scandinavia suggest that this oilseed was an important oil crop 2000 years ago [37] and the interest in CA has now been renewed in some parts of Central and Northern Europe as a rich source of the essential *n*-3 fatty acid ALA [38].

Many fatty fish species are, in general, good sources of the healthy *n*-3 fatty acids EPA and DHA for human consumption, but the fatty acid composition varies between species. The North Atlantic fish species capelin, herring, and sandeel contain lower levels of EPA and DHA compared to the South American sardine, 8% compared to 23% of total fatty acids, respectively [39]. In addition, oils from North Atlantic fish spices are richer in the long-chain monounsaturated fatty acids (LC-MUFAs), in particular, cetoleic acid (22:1 *n*-11) with 17–22% compared to 1% in sardine [39]. Compared to EPA and DHA, relatively little is known about the physiological functions and health impact of different LC-MUFAs found in fish oils. Recently, we showed that cetoleic acid improves the conversion of ALA to EPA and DHA in two different in vitro cell model systems, primary salmon liver cells and human HepG2 cell line, and in vivo in a feeding trial with salmon [40]. These findings suggest the hypothesis that utilisation of the plant-based omega-3 fatty acid ALA to EPA and DHA can be increased by combining dietary oils, one rich in ALA and the other rich in cetoleic acid. Thus, the aim of the present study was to investigate the effects of diets containing CA with a high content of ALA, alone or in blends with sandeel oil (SA) high in cetoleic acid, on the uptake and utilisation of ALA and its conversion to EPA and DHA, as well as the effects on gene expression of enzymes of the *n*-3 fatty acid pathway in Zucker fa/fa rats. Different ratios of CA and SA were selected to investigate if there are any dose-response effects of increasing ALA and cetoleic acid, respectively.

## 2. Materials and Methods

### 2.1. Animals, Diets, and Feeding Trial

The animal experiment was approved in accordance with the Norwegian regulation on animal experimentation (approval no. 11603). The Norwegian State Board of Biological Experiments with Living Animals approved the protocol. Soybean oil was delivered from Dyets Inc. (Bethlehem, PA, USA). The fish oil (SA) was sandeel oil produced by Vedde AS (Langevåg, Norway). Cold-pressed camelina oil (CA) was produced by Norsk Matraps SA at Askim Frukt og Bærpresseri (Askim, Norway). Casein was purchased from Sigma-Aldrich (Saint-Louis, MO, USA). All other feed ingredients were purchased from Dyets Inc. (Bethlehem, PA, USA).

Thirty-six male Zucker fa/fa rats (Crl:ZUC(Orl)-Lepr fa, from Charles River Laboratories, Italy) were assigned to six experimental groups of six rats each with comparable mean body weight. The rats were housed in pairs in individually ventilated cages (IVC type 4, blue line from Tecniplast, Italy) in a room maintained at a 12 h light–dark cycle and a constant temperature of 21 ± 1 °C. The rats were acclimatised under these conditions before the start of the experiment. The intervention period started when the rats were 10–12 weeks old and weighed 426 ± 16 g. The rats were fed a modified, semi-purified experimental diet based on AIN-93G recommendation for growing rats [41] (Table 1). All the diets contained 12 wt% fat. The control diet (Ctrl) contained 12% soybean oil while the other diets contained 7% soybean oil and 5% experimental oil. The experimental oil contained either 100% CA, 75% CA, and 25% SA (CA75 + SA25); 50% CA and 50% SA (CA50 + SA50); 25% CA and 75% SA (CA25 + SA75); or 100% SA (SA). The diets were prepared according to the instructions for the AIN-93G [41], and as recommended by Dyets Inc 1 wt% growth and maintenance supplement was added to all diets. Since the feed was given as a powder formula, the rats always had access to wood chewing sticks. The rats were fed these diets for four weeks and were weighed weekly during the intervention period. One rat in the CA50 + SA50 group had to be euthanised in the third week of the study due to a serious lesion and is not included in the results; therefore, n = 5 in this group and n = 6 in all other groups. At the end of the feeding period (after 4 weeks of intervention), under non-fasting conditions with free access to feed and tap water, the rats were euthanised while under anaesthesia with Isofluran (Isoba vet, Intervet, Schering–Plough Animal Health, Boxmeer, The Netherlands) mixed with nitrous oxide and oxygen. Blood was drawn directly from the heart and was collected in Vacuette Z Serum Clot Activator Tubes (Greiner Bio-one, Kremsmünster, Austria) for isolation of serum and in Vacuette 3 mL K2EDTA tubes (Greiner Bio-one) for isolation of plasma. The blood cells (rest of the blood fraction after removal of plasma), liver, and heart were frozen and stored at −80 °C until analysis. 

### 2.2. Analyses in Serum and Plasma

Total cholesterol, HDL-cholesterol, triacylglycerol, alanine transaminase, aspartate transaminase, non-esterified fatty acids (NEFA), and total bile acids in serum were analysed on the Cobas c111 system (Roche Diagnostics GmbH, Mannheim, Germany) using the appropriate kits from Roche Diagnostics. The NEFA FS kit (DiaSys Diagnostics Systems GmbH, Holzheim, Germany) and the Bile Acid Assay kit (Diazyme Laboratories, Inc., Poway, CA, USA) were used. Serum alanine transaminase (ALAT) and aspartate transaminase (ASAT) were measured with pyridoxal phosphate activation. C-reactive protein (CRP) was measured in plasma using the Rat C-Reactive Protein ELISA (cat.no. 88-7501, from Invitrogen by Thermo Fisher Scientific, Waltham, MA, USA).

### 2.3. Extraction of Total Lipids

Total lipids were extracted from diets, blood cells, liver, and heart as described by Folch et al. [42].

### 2.4. Analyses of Fatty Acid Composition in Diets and Blood Cells

Fatty acid methyl esters (FAME) in total lipids from diets and blood cells were prepared by acid-catalysed transesterification of total lipids according to the method of Christie et al. [43]. Extraction and purification of FAME were performed as described by Ghioni et al. [44]. FAME were separated by gas–liquid chromatography using a Thermo Fisher Trace GC 2000 (Thermo Fisher, Hemel Hempstead, UK) equipped with a fused silica capillary column (ZBWax, 60 m × 0.32 × 0.25 mm i.d.: Phenomenex, Macclesfield, UK) with hydrogen as carrier gas and using on-column injection.

### 2.5. Analyses of Fat Content and Fatty Acid Composition in Liver and Heart

After the extraction of lipids from the liver and heart, a known volume of the chloroform extract was put into a previously weighed beaker. The chloroform was evaporated, and the fat content was weighed. Then, the amount of fat in % in the sample was calculated in an equation taking into consideration the ratio of distribution of chloroform in the different phases in the extraction. FAME in total lipids from liver and heart were made according to a method described by Mason et al. [45] and Hoshi et al. [46]. The FAME were separated by gas–liquid chromatography using a Hewlett Packard 6890 GC with HP ChemStation software (Chemstation, OpenLab 2010–2014) equipped with a fused silica capillary column (BPX70, 60 m × id 0.25 mm and a film thickness of 0.25 μm; SGE Analytical Science™, now Trajan Scientific and Medical, Victoria, Australia) with helium as carrier gas and using a split injector. Injector and detector temperatures were both 270 °C. The temperature was raised from 50 °C to 170 °C at a rate of 4 °C /min, then raised to 200 °C at a rate of 0.5 °C/min and, finally, to 240 °C at a rate of 10 °C/min, with a hold time for 12 min. The relative quantity of each fatty acid was determined by measuring the area under the peak in the GC spectrum corresponding to that fatty acid.

### 2.6. Analyses of Gene Expression

Total RNA was isolated from the liver using PureLink™ Pro 96 RNA Purification Kit with Dnase I treatment (Invitrogen, Carlsbad, CA, USA) following the producers’ protocol. Concentration and purity of RNA were evaluated using NanoDrop 1000 Spectrophotometer ((NanoDrop Technologies, Wilmington, DE, USA)). cDNA was synthesised from 1000 ng RNA in a 20 μL reaction volume by using Taqman reverse transcriptase reagents (Applied Biosystems, Foster City, CA, USA). The cDNA synthesis was run under the following conditions: 25 °C for 10 min, 37 °C for 30 min to synthesise cDNA, and 95 °C for 5 min to terminate the cDNA synthesis reaction. The reaction mix of qPCR consisted of 4 μL diluted (1:7.5) cDNA, 1 μL forward and reverse primer (final concentration of 0.5 μM, Appendix A), and 5 μL PowerUp SYBR Green Master Mix (Applied Biosystems, Foster City, CA, USA). A standard curve was included for each primer pair to evaluate the primer efficiency. All samples were analysed in parallel, and a non-template control with water substituted for cDNA was run for each primer pair. The qPCR reaction was run on a QuantStudio 5 instrument (Thermo Fisher Scientific, Waltham, MA, USA) under the following conditions: Initial denaturation 95 °C for 20 s, amplification with 40 cycles at 95 °C for 1 s and 60 °C for 20 s, melting at 95 °C for 1 s and 60 °C for 20 s, dissociation 95 °C for 1 s. *Eif3a*, *Rpol2*, and *Ef1a* were evaluated using RefFinder [47]. The relative gene expression level was calculated according to the ΔΔCt method [48] using Eif3a as a reference gene.

### 2.7. Statistics

Statistical analyses were conducted using the software package UNISTAT (London, UK) and SAS^®^ System for Windows Release 9.4 (SAS Institute Inc., Cary, NC, USA). The fatty acid profile in blood cells, liver, and heart, gene expression, and serum/plasma data were subjected to multiple comparisons using one-way analysis of variance (ANOVA). Statistical units were individuals per dietary group (n = 6, except for CA50 + SA50 which had n = 5). Significant effects were indicated at a 5% level. The differences were ranked using the Tukey-HSD post hoc test. Correlations between dietary and tissue content of fatty acids were evaluated by Pearson’s correlation coefficients.

## 3. Results

### 3.1. Fat, Energy, and Fatty Acid Composition of the Diets

The total fat or total energy between the diets were comparable (Table 1), but the fatty acid composition differed (Table 2). The main fatty acids in the Ctrl diet with 120 g of soybean oil/kg feed were linoleic acid (LA; 18:2*n*-6) (55.0% (*w*/*w*)), oleic acid (18:1*n*-9) (19.8% (*w*/*w*)), and palmitic acid (16:0) (11.1% (*w*/*w*)). Due to the addition of 50 g soybean oil/kg feed in all the experimental diets, these three fatty acids were also the main fatty acids of the other experimental diets with CA and/or SA. The level of LA, the only *n*-6 fatty acid in the diets, decreased from 55% (*w*/*w*) in the Ctrl diet to 24.8% (*w*/*w*) in the SA diet. The only *n*-3 fatty acid in the Ctrl diet and CA diet was ALA, which made up 6.9% (*w*/*w*) and 25.3% (*w*/*w*) of the total fatty acids, respectively. The SA diet contained total 18.3% (*w*/*w*) *n*-3 fatty acids including 3.7%, (*w*/*w*) ALA, 6.0% (*w*/*w*) EPA, and 5.3% (*w*/*w*) DHA. The total content of MUFAs in the diets differed from 21.7% (w/w) in the Ctrl diet to 32.9% (*w*/*w*) in the SA diet with oleic acid (18:1 *n*-9) as the main MUFA. The Ctrl diet and the CA diet did not contain cetoleic acid (22:1 *n*-11), while the SA diet contained 8.7% (*w*/*w*) cetoleic acid. The three diets CA75 + SA25, CA50 + SA50, and CA25 + SA75 contained 50 g soybean oil and 70 g of different blends of SA and CA/kg feed. There is a linear decrease in the content of ALA with increased SA and decreased CA in the diets (R^2^ = 0.9992). In parallel, the content of DHA, DPA, EPA, and cetoleic acid content in these diets decreased (R^2^ = 0.998–0.999) with SA > 25CA + 75SA < 50CA + 50SA < 75CA + 25SA < CA.

### 3.2. Biometric Data

During the four-week feeding trial, the body weight of the rats increased from a start weight of 426 ± 16 g to a final weight of 519 ± 25 g. The dietary groups were similar with regard to the body weight at endpoint (Appendix A).

### 3.3. Plasma and Serum Parameters

There were no significant differences between the groups in the plasma or serum parameters analysed (Appendix A).

### 3.4. Fat Content in the Liver and Heart

The average fat content in the liver was 12.9 ± 4.9% and in the heart 4.6 ± 1.9% for all groups (Appendix A) with no significant differences between the groups.

### 3.5. Fatty Acid Composition

The different diets significantly affected the fatty acid composition of blood cells, liver, and heart, showing to a large extent a reflection of the dietary fatty acid composition with a few exceptions (Figure 1, Figure 2, Figure 3 and Figure 4, Appendix A).

#### 3.5.1. Blood Cells

The amounts of *n*-3 fatty acids in blood cells differed from 3.70% (*w*/*w*) to 11.28% (*w*/*w*), with a significantly lower level in the Ctrl group compared to the other groups (Figure 1). ALA varied from 0.44% (Ctrl) to 2.21% (CA) of total fatty acids, with increasing CA inclusion in the diet resulting in increasing ALA in blood cells. The amounts of EPA, DPA, and DHA in blood cells were higher (15.8, 2.5, and 2.8 times, respectively) in the SA group compared to the Ctrl group, whereas DPA in blood cells was less affected by varying CA and SA content in the diets; EPA and DHA increased by increasing SA in the diets. The amounts of DPA and DHA were significantly higher in blood cells from the CA group given camelina oil with high amounts of ALA, compared with the Ctrl group given only soybean oil with a low amount of ALA. EPA and DHA were significantly higher in the CA25 + SA75 and the SA groups, compared to the other groups (Figure 1). The *n*-6/*n*-3 ratio decreases significantly from 8.31 in the Ctrl group, with only soybean oil in the diet, to 3.42 in the CA group with a mix of soybean oil and camelina oil in the diet (Figure 4). Compared to the CA group, the *n*-6/*n*-3 ratio was lower in the CA25 + 75SA and SA groups; however, this was not the case when compared to CA75 + SA25 and CA50 + SA50. No significant differences in the total amounts of saturated fatty acids (SFAs), MUFAs, or *n*-6 PUFAs were observed. In the blood cells from the Ctrl group, with only soybean oil in the diet, a significantly lower amount of 20:1*n*-9 and significantly higher amounts of the three *n*-6 fatty acids (18:2, 22:4 and 22:5) were observed. Cetoleic acid from the dietary sandeel oil was not detected in the blood cells.

#### 3.5.2. Liver

The amount of total *n*-3 fatty acids in the liver increased significantly from 3.1% in the Ctrl group to 9.1% of the fatty acids in the SA group (Figure 2). The *n*-6/*n*-3 ratio was 5.9 in the Ctrl group, and significantly higher than in all the other groups (Figure 4). The amount of ALA varied significantly from 0.4% (*w*/*w*) (Ctrl) to 1.8% (*w*/*w*) (CA) of total fatty acids, and increasing CA inclusion in the diet results in increasing ALA in the liver (CA > CA75 + SA25 > CA50 + SA50 > SA). No difference in EPA, DPA, and DHA levels between the groups fed different inclusions of CA was observed. However, the SA group showed significantly higher EPA and DPA than the other groups. The amounts of EPA, DPA, and DHA in the liver were highest in the SA group and lowest in the Ctrl group (15.0, 5.0, and 2.2 times higher, respectively). No significant difference in total SFAs, MUFAs, or *n*-6 PUFAs was observed between the groups. Cetoleic acid from the dietary sandeel oil was not detected in the liver samples.

#### 3.5.3. Heart

The total amount of *n*-3 fatty acids in the heart varied from 11.1% (*w*/*w*) in the Ctrl group with only soybean oil in the diet to 16.7% (*w*/*w*) in the CA group and 20.1% (*w*/*w*) in the SA group (Figure 3). The *n*-6/*n*-3 ratio is 3.4 in the Ctrl group, which is significantly higher than in the other groups with a ratio between 1.3 (SA) and 1.9 (CA) (Figure 4). The amount of ALA varied from 0.4% (*w*/*w*) (Ctrl) to 1.4% (*w*/*w*) (CA75 + SA25) of total fatty acids. EPA and DHA levels increased by increasing SA in the diet, whereas DPA showed a different trend with higher levels in the CA compared to all the other groups. Both CA25 + SA75 and SA had a significantly higher amount of EPA than the other groups, while the amount of DHA was only significantly higher in SA compared to Ctrl, CA, and CA75 + SA25. The Ctrl group had 37.1% (*w*/*w*) *n*-6 fatty acids in the hearts, which is a significantly higher amount than in all the other groups, mainly due to higher amounts of the three *n*-6 fatty acids 20:4, 22:4, and 22:5. No significant difference in the total amounts of SFAs or MUFAs were observed, but the marine MUFA cetoleic acid (22:1*n*-11) was observed in the hearts of the groups fed ≥50% SA, with the highest amount in the SA group (0.7% (*w*/*w*)). No erucic acid (22:1*n*-9) from the CA or SA was observed in the heart. 

### 3.6. Gene Expression of Enzymes of the n-3 Fatty Acid Pathway in the Liver

The liver gene expression was significantly affected by the dietary content of CA and SA (Figure 5a–c); the higher dietary content of SA induced a strong trend towards lower liver gene expression of both *Elovl5, Fads1*, and *Fads2*. There were no significant differences (*p* < 0.05) in gene expression of *Elovl5, Fads1*, and *Fads2* between Ctrl and the groups fed the two lowest dietary ratios of CA; CA and CA75 + SA25; however, a trend towards a lower expression of *Elov5* and *Fads1* in CA75 + SA25 compared to Ctrl was observed. Rats fed the diet containing SA showed the lowest gene expression levels of *Fads1*.

## 4. Discussion

In the WHO European Region, more and more people are shifting towards plant-based diets for reasons relating to health, as well as to ethical considerations about climate change and animal welfare. In some countries, changes in dietary patterns are only just emerging, while in others this trend is increasing rapidly. Nevertheless, the evidence for the long-term health implications of vegetarian and vegan diets remains incomplete [49]. In a more plant-based diet, ALA will be the main *n*-3 fatty acid and the health effects of ALA as the main *n*-3 fatty acid are still uncertain. The conversion of ALA to the health-documented EPA and DHA by the *n*-3 pathway is known, but the capacity and possible inducers of these *n*-3 pathways are discussed.

In the present trial with obese Zucker fa/fa rats, the *n*-3 fatty acids EPA, DPA, and DHA were detected in both blood cells, liver, and heart of all groups, even the groups (Ctrl and CA) that were not given these long chain *n*-3 fatty acids through the diets (Table 2, Figure 1, Figure 2 and Figure 3). On the other hand, the Ctrl and CA diets contained ALA, and it is known that EPA and DHA can be endogenously synthesised through multiple steps from ALA through a well-conserved *n*-3 pathway [8] and with the same enzymes identified in many species (humans [9], rats [10], salmon [11,12]). Our results demonstrate a certain capacity for both EPA and DHA synthesis from ALA in the obese Zucker fa/fa rat model when they were given an increased amount of ALA from CA in the diet. This is in accordance with Hong et al. who reported elevated levels of ALA, EPA, DPA, and DHA in liver TG and PL in Zucker rats after a flaxseed-based diet high in ALA [50]. It has been reported that the conversion of ALA to EPA and DHA is limited in many species and seems to differ between female and male. In humans, the conversion capacity of ALA to DHA is estimated to be <10% in females, whereas the capacity in males is only <3% [6,7]. Only male rats were included in the present study. The lower capacity in males compared to females previously observed in humans, might indicate a potential for improvement of EPA and DHA synthesis in males. Several rat and human studies have reported increased EPA content in plasma phospholipid with increased ALA intake, concomitant with no change in plasma DHA [4,17]. Human HepG2 cells also showed a higher capacity for synthesis of EPA compared to DHA when the cells were cultured in growth media enriched with cetoleic acid [40]. In addition, lower DHA than EPA status is observed in humans on vegetarian or vegan diets [5], indicating a low utilisation of ALA being converted to DHA. Another study on rats indicated that endogenous DHA synthesis from ALA may be sufficient to supply the brain with this important fatty acid [4]. Increased levels of ALA in the diets can improve conversion to the longer-chain *n*-3 fatty acids in rats [51] and humans [52]. A positive effect of the almost four times higher dietary ALA levels in the CA than in the Ctrl is shown by the 4.2–4.5 times higher EPA, 1.8–2.6 times higher DPA, and 1.3–1.5 times higher DHA in the blood cells, liver, and heart of the CA than the Ctrl. Compared to the Ctrl, the CA group showed significantly higher DPA (blood cells and heart) and DHA (blood cells). Conversion of ALA to longer chain *n*-3 fatty acids has been confirmed in other studies of rats [24,25,26,27,28,53].

The groups fed different blends of CA and SA showed a trend towards increasing EPA and DHA in the blood cells, liver, and heart by increasing SA in the diets. However, in blood cells, no significant differences in levels of EPA and DHA between the SA group and the CA25 + SA75 group were observed. Thus, 25% of the fish oil in the diet could be replaced with a plant oil high in ALA without any significant change in levels of EPA and DHA in blood cells. Increasing in vivo content of EPA and DHA can partially be explained by the content of these fatty acids in the diets. In addition, the content of the bioactive cetoleic acid was increased by the inclusion level of SA in the diets, and can also explain a higher levels of EPA, DPA and DHA in the rats. In vivo and in vitro studies have shown that cetoleic improves the EPA and DHA synthesis in salmon and human liver cells and increases deposition of these fatty acids in salmon muscle [40]. In the present study, the EPA level in the blood cells showed a moderate negative correlation with the dietary ALA. This, in addition to a strong positive correlation between dietary cetoleic acid (from the fish oil) and the level of EPA (0.931, *p* < 0.0001), DPA (0.684, *p* < 0.0001), and DHA (0.882, *p* < 0.0001) in the blood cells, supports an increased conversion of ALA to both EPA and DHA by dietary cetoleic acid in this study (Table 3).

The synthesis of EPA and DHA from ALA requires several elongation and desaturation steps, involving the enzymes fatty acid desaturase 1 and 2 and elongase 2 and 5 [13,14,15,16]. In our study, there were no significant differences in gene expression of *Elovl5, Fads1*, and *Fads2* between the Ctrl and the CA group, showing no effect of increased ALA in the diet on the gene expression. This is in line with Tu et al. [54], showing no difference in gene expression of *Fads1*, *2* and *Elovl2*,5 in rats when ALA was changed from 0.2 to 2.9 percent of energy. The liver gene expression in obese Zucker fa/fa rats was significantly affected by the dietary content of the combinations of CA and SA, with a tendency of reduced gene expression of *Fads1*, *2* and *Elovl5* by increasing SA. Several studies have shown that dietary fatty acids (such as lipoic acid and cetoleic acid) may exert a great influence on the efficiency of the *n*-3 fatty acid pathway [32,40,55,56,57], but the *n*-3 fatty acid synthesis might be regulated more by substrate levels than by gene expression [54]. 

The present results show increases or trends towards increased EPA, DPA, and DHA in both blood cells, liver, and heart. However, the intake of ALA has been reported to increase EPA content in plasma phospholipid, but not increased DHA in several rat and human studies [4,17], and this may be due to a rate-limiting step between EPA and DHA in the *n*-3 fatty acid pathway [58]. It has been reported that high dietary levels of DHA inhibit the *n*-3 metabolic pathway in different species by feedback inhibition [27,29,30,31], whereas lack of DHA in the diet stimulates the *n*-3 fatty acid biosynthetic pathway [32,33,34]. Other factors shown to influence the capacity for EPA and DHA synthesis include diet composition, life stage, genotype, and growth [18,19,20,21,22,23]. In our study, the dietary composition of the *n*-3 fatty acids was to a large degree reflected in blood cells, liver, and heart, showed by strong positive correlations (Table 3). This has also been reported in several other studies [32,33]. However, a comparison of DHA content in the diets versus in the liver shows an increase in DHA deposition in the liver at relatively low dietary SA inclusion (25%) (Figure 6). Higher SA content (50% and 75%) resulted in a drop or minor change in liver DHA, whereas the high dietary SA (100%) further increased the liver DHA. This increase might be caused by a stimulatory effect of high concentrations of cetoleic acid in the fish oil and is in accordance with our earlier study on cetoleic acid and ALA in salmon showing that high, and not low cetoleic acid levels, stimulate the *n*-3 pathway [40].

It is unknown if cetoleic acid is the only monounsaturated fatty acid that stimulates this pathway. In addition to a high amount of ALA, CA contains a small amount of erucic acid (22:1 *n*-9) and a high amount of monounsaturated fatty acid with 20 carbons in the chain (20:1 *n*-9). In rats, high dietary levels of erucic acid have been associated with myocardial lipidosis [59]. In our study, the dietary level of erucic acid was low, erucic acid was not detected in the hearts, and no difference in lipid level in the heart was observed. It is still possible that the high amount of 20:1 *n*-9 in camelina oil may have a stimulatory effect on the *n*-3 pathway, such as that of cetoleic acid, but this must be further investigated.

## 5. Conclusions

In conclusion, the present study shows significantly higher levels of the long-chain *n*-3 fatty acids DPA and DHA in blood cells from obese Zucker fa/fa rats given CA with high amounts of ALA compared with the rats given only soybean oil with a low amount of ALA (Ctrl). This indicates an active conversion of ALA to the longer *n*-3 fatty acids DPA and DHA in these rats when given a high amount of ALA and no EPA, DPA, and DHA in the feed. In addition, the results show an increase in the uptake and deposition of EPA and DHA along with an increase in the dietary content of SA. However, 25% of the SA could be exchanged with CA without having a significant effect on EPA, DPA, or DHA in blood cells, indicating that bioactive components in SA, as cetoleic acid, might counteract the inhibiting effect of the high dietary content of DHA on the *n*-3 biosynthetic pathway.

## Figures and Tables

**Figure 1 nutrients-15-02344-f001:**
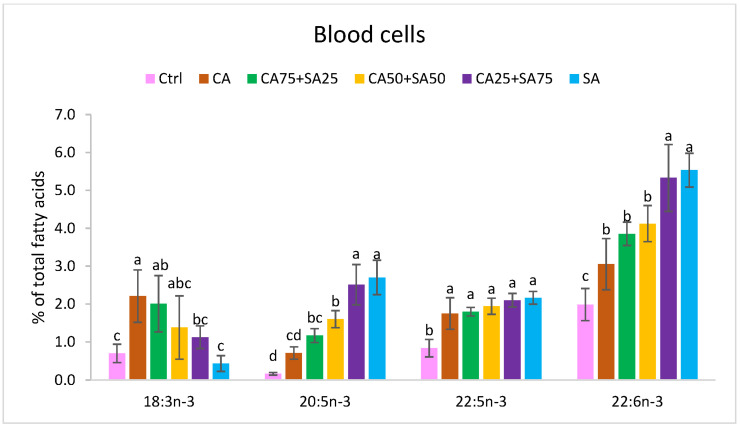
Content (% of total fatty acids) of 18:3 *n*-3, 20:5 *n*-3, 22:5 *n*-3, and 22:6 *n*-3 in blood cells. Data are means (n = 6, except for CA50 + SA50 which had n = 5) shown with standard deviation. Different letters indicate significant differences (*p* < 0.05) evaluated using one-way ANOVA followed by Tukey’s HSD. Ctrl = Control, CA = Camelina oil, SA = Sandeel oil.

**Figure 2 nutrients-15-02344-f002:**
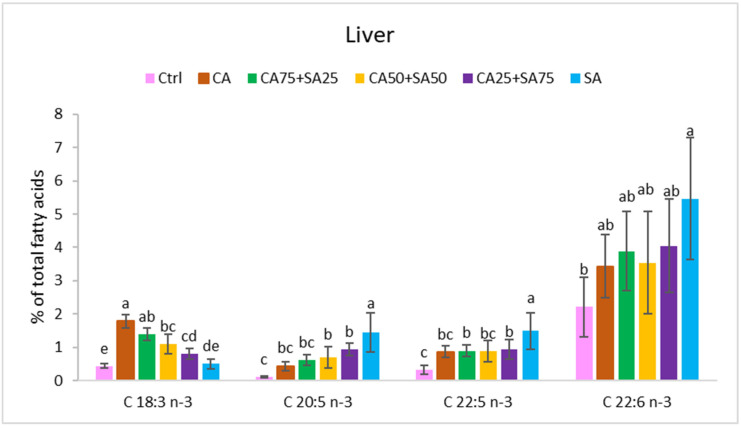
Content (% of total fatty acids) of 18:3 *n*-3, 20:5 *n*-3, 22:5 *n*-3 and 22:6 *n*-3 in the liver. Data are means (n = 6, except for CA50 + SA50 which had n = 5) shown with standard deviation. Different letters indicate significant differences (*p* < 0.05) evaluated using one-way ANOVA followed by Tukey’s HSD. Ctrl = Control, CA = Camelina oil, SA = Sandeel oil.

**Figure 3 nutrients-15-02344-f003:**
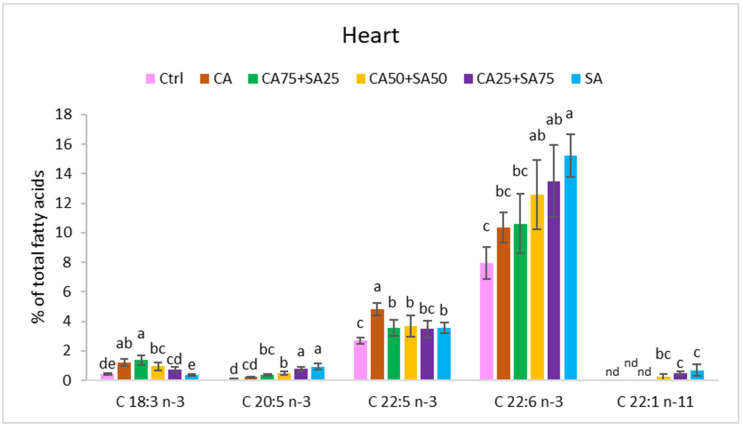
Content (% of total fatty acids) of 18:3 *n*-3, 20:5 *n*-3, 22:5 *n*-3, 22:6 *n*-3 and 22:1 *n*-11 in heart. Data are means (n = 6, except for CA50 + SA50 which had n = 5) shown with standard deviation. Different letters indicate significant differences (*p* < 0.05) evaluated using one-way ANOVA followed by Tukey’s HSD. Ctrl = Control, CA = Camelina oil, SA = Sandeel oil.

**Figure 4 nutrients-15-02344-f004:**
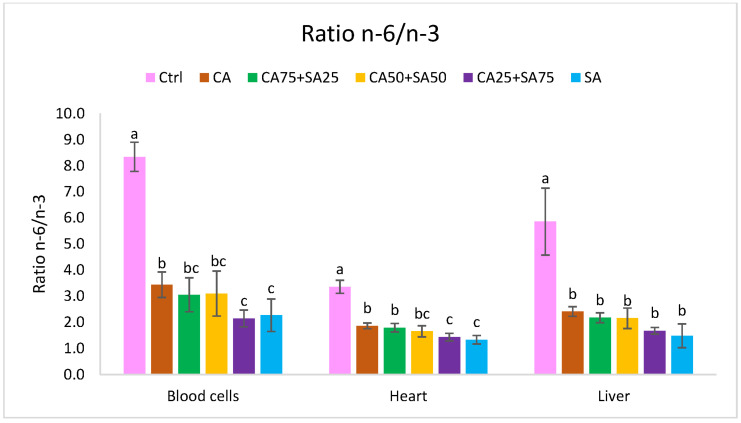
*n*-6/*n*-3 ratio in blood cells, heart, and liver. Data are means (n = 6, except for CA50 + SA50 which had n = 5) shown with standard deviation. Different letters indicate significant differences (*p* < 0.05) evaluated using one-way ANOVA followed by Tukey’s HSD. Ctrl = Control, CA = Camelina oil, SA = Sandeel oil.

**Figure 5 nutrients-15-02344-f005:**
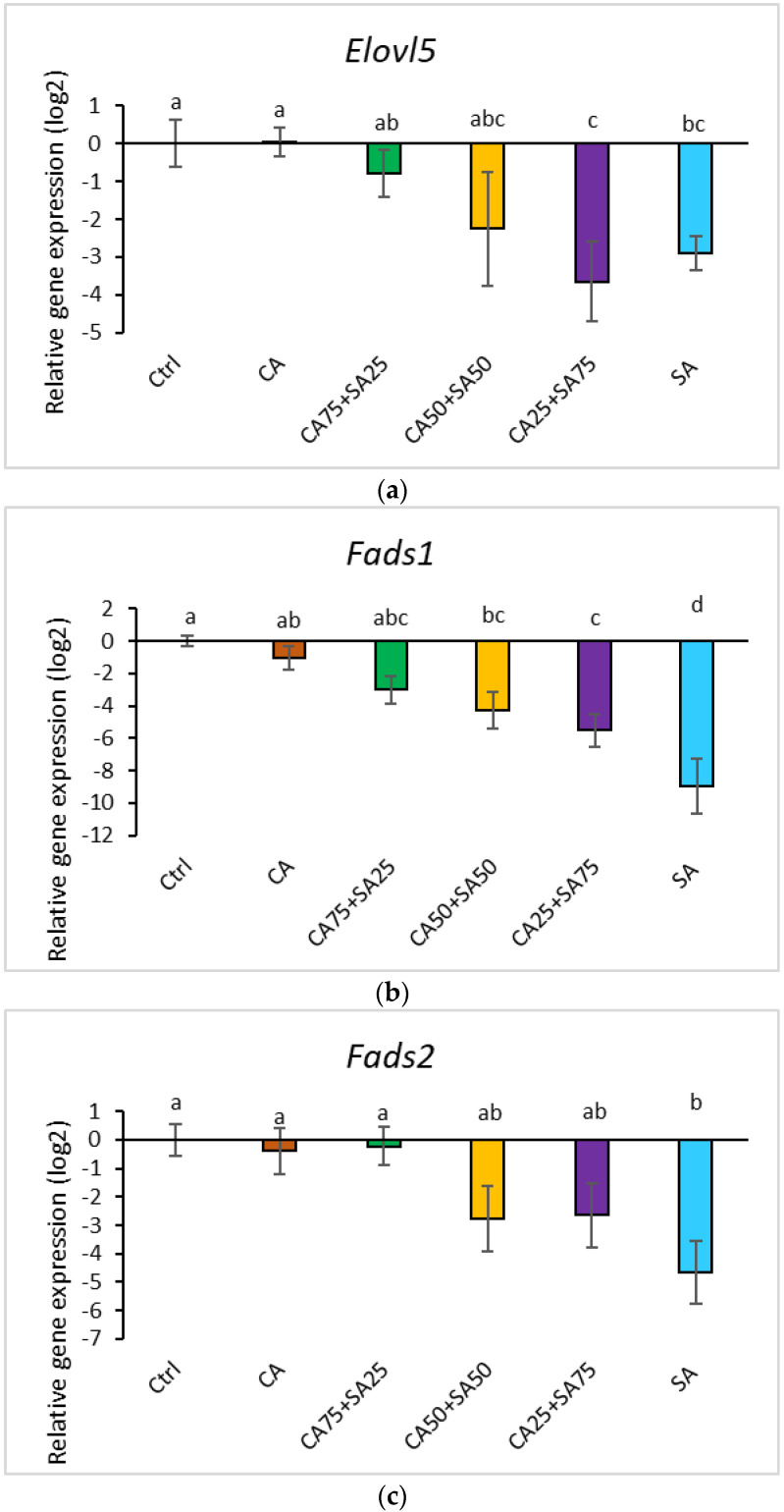
Relative gene expression of (**a**) *Elovl5*, (**b**) *Fads1*, and (**c**) *Fads2* in liver. Data are means (n = 6, except for CA50 + SA50 which had n = 5) shown with standard error mean. Different letters indicate significant differences (*p* < 0.05) evaluated using one-way ANOVA followed by Tukey’s HSD. Ctrl = Control, CA = Camelina oil, SA = Sandeel oil. *Elovl5* = fatty acid elongase 5, *Fads1* = fatty acid desaturase 1, *Fads2* = Fatty acid desaturase 2.

**Figure 6 nutrients-15-02344-f006:**
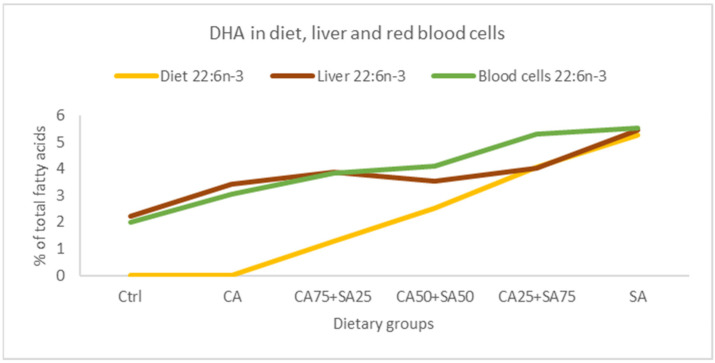
Comparison of DHA (22:6*n*-3, % of total fatty acids) in the diet, in the liver, and in the blood cells from the different dietary groups. Data are means (n = 6, except for CA50 + SA50 which had n = 5). Ctrl = Control, CA = Camelina oil, SA = Sandeel oil.

**Table 1 nutrients-15-02344-t001:** Composition (g/kg diet), fat (g/100 g diet), and energy content (kJ/g diet) of the experimental diets. Ctrl = control, CA = Camelina oil, SA = Sandeel oil.

	Ctrl	CA	CA75 + FO25	CA50 + SA50	CA25 + SA75	SA
Soybean Oil ^1^	120	50	50	50	50	50
Sandeel Oil ^2^	-	-	17.5	35	52.5	70
Camelina Oil ^3^	-	70	52.5	35	17.5	-
Casein protein ^4^	232	232	232	232	232	232
Corn starch ^5^	446	446	446	446	446	446
Sucrose ^5^	90	90	90	90	90	90
Cellulose ^5^	50	50	50	50	50	50
Tert-butylhydroquinone ^5^	0.014	0.014	0.014	0.014	0.014	0.014
Mineral Mix (AIN-93MX) ^5^	35	35	35	35	35	35
Vitamin Mix (AIN-93VX) ^5^	10	10	10	10	10	10
L-Methionine ^5^	1.6	1.6	1.6	1.6	1.6	1.6
L-Cystine ^5^	3	3	3	3	3	3
Choline Bitartrate ^6^	2.5	2.5	2.5	2.5	2.5	2.5
Growth and Maintenance Supplement ^7^	10	10	10	10	10	10
Energy content (kJ/g diet)	19.8	19.5	19.8	19.8	19.8	19.8
Total fat (g/100 g diet)	11.3	11	10.7	11.2	10.8	11.3

^1^ Dyets Inc. (Bethlehem, PA, USA). ^2^ Vedde, Langevåg, Norway. ^3^ Norsk Matraps at Askim Frukt og Bærpresseri (Askim, Norway). ^4^ Sigma-Aldrich (Saint-Louis, MO, USA), contains 86.25% crude protein. ^5^ Dyets Inc. (Bethlehem, PA, USA). ^6^ Dyets Inc. (Bethlehem, PA, USA), contains 41.1% choline. ^7^ Dyets Inc. (Bethlehem, PA, USA), contains 2.5 g vitamin B12, 2.5 g vitamin K1, and 995 g sucrose per kg.

**Table 2 nutrients-15-02344-t002:** Fatty acid composition (*w*/*w*, % of total fatty acids) of the experimental diets. Ctrl = control, CA = Camelina oil, SA = Sandeel oil, SFA = saturated fatty acids, MUFA = monounsaturated fatty acids, PUFA = polyunsaturated fatty acids.

	Ctrl	CA	CA75 + SA25	CA50 + SA50	CA25 + SA75	SA
C14:0	0.1	0.1	0.9	1.8	2.8	3.7
C16:0	11.1	8.1	9.1	10.3	11.5	12.6
C18:0	4.1	3.4	3.2	3.1	2.9	2.8
C20:0	0.4	0.7	0.6	0.5	0.4	0.3
Σ SFA	15.7	12.2	13.9	15.6	17.5	19.3
C16:1 *n*-7	0.1	0.1	1.0	1.9	3.0	3.8
C18:1 *n*-9	19.8	17.5	16.3	15.1	13.5	12.4
C18:1 *n*-7	1.5	1.2	1.3	1.4	1.5	1.6
C20:1 *n*-9	0.3	6.0	6.1	5.9	6.0	5.7
C22:1 *n*-11	0.0	0.0	2.1	4.2	6.7	8.7
C22:1 *n*-9	0.0	0.7	0.7	0.7	0.7	0.7
Σ MUFA	21.7	25.6	27.4	29.1	31.4	32.9
C18:2 *n*-6	55.0	34.2	31.6	29.5	26.4	24.8
Σ *n*-6 PUFA	55.0	34.2	31.6	29.5	26.4	24.8
C18:3 *n*-3	6.9	25.3	20.3	15.0	9.3	3.7
C18:4 *n*-3	0.0	0.0	0.7	1.2	2.0	2.5
C20:5 *n*-3	0.0	0.0	1.5	2.9	4.7	6.0
C21:5 *n*-3	0.0	0.0	0.1	0.2	0.3	0.4
C22:5 *n*-3	0.0	0.0	0.1	0.2	0.4	0.5
C22:6 *n*-3	0.0	0.0	1.3	2.5	4.1	5.3
Σ *n*-3 PUFA	6.9	25.3	23.9	22.1	20.7	18.3
*n*-6/*n*-3	8.0	1.4	1.3	1.3	1.3	1.4

**Table 3 nutrients-15-02344-t003:** Correlations between dietary fatty acids and omega-3 fatty acid composition in blood, liver, and heart. *p* values in brackets.

		Dietary Fatty Acids				
		18:3*n*-3	20:5*n*-3	22:5*n*-3	22:6*n*-3	22:1*n*-11
Blood	18:3*n*-3	0.796 (<0.0001)	-	-	-	-
	20:5*n*-3		0.932 (<0.0001)	0.931 (<0.0001)	0.932 (<0.0001)	0.931 (<0.0001)
	22:5*n*-3	-	0.686 (<0.0001)	0.687 (<0.0001)	0.686 (<0.0001)	0.684 (<0.0001)
	22:6*n*-3	-	0.883 (<0.0001)	0.882 (<0.0001)	0.883 (<0.0001)	0.882 (<0.0001)
Liver	18:3*n*-3	0.931 (<0.0001)	-	-	-	
	20:3*n*-3	0.777 (<0.0001)	-	-	-	
	20:5*n*-3	-	0.799 (<0.0001)	0.800 (<0.0001)	0.801 (<0.0001)	0.801 (<0.0001)
	22:5*n*-3	-	0.623 (<0.0001)	0.626 (<0.0001)	0.625 (<0.0001)	0.626 (<0.0001)
	22:6*n*-3	-	-	-	-	-
Heart	18:3*n*-3	0.838 (<0.0001)	-	-	-	-
	20:3*n*-3	0.611 (<0.0001)	-	-	-	-
	20:5*n*-3	-	0.920 (<0.0001)	0.918 (<0.0001)	0.920 (<0.0001)	0.920 (<0.0001)
	22:5*n*-3	0.613 (<0.0001)	-	-	-	-
	22:6*n*-3	-	0.785 (<0.0001)	0.787 (<0.0001)	0.786 (<0.0001)	0.786 (<0.0001)

## Data Availability

Not applicable.

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
