# Peer review of "Different Dietary Ratios of Camelina Oil to Sandeel Oil Influence the Capacity to Synthesise and Deposit EPA and DHA in Zucker Fa/Fa Rats"

_nutrients, 2023, doi:10.3390/nu15102344_

Round 1

Reviewer 1 Report

This work is devoted to the study of the effect of various ratios of camelina oil and sandstone oil on the ability to synthesize and deposit EPA and DHA in rats. The article is written in understandable language. The main points are beyond doubt. The abundance of data, subject matter, and length of the manuscript meet the requirements of the journal. The article is well structured and makes a good impression. There are some points for improvement:

1. It is not entirely clear from the text why the authors took precisely such ratios of components.

2. Results for camelina oil and sandstone oil in various ratios should be compared with mixtures of other oils in more detail.

3. It is desirable to indicate for comparison what other factors and substances can influence the synthesis and deposition of EPA and DHA in rats.

4. Please cite: 10.3390/molecules27186129.

5. The article has a voluminous part "Discussion", but there is no "Conclusions". It is desirable to move some generalizations and main results to "Conclusions".

Author Response

  1. It is not entirely clear from the text why the authors took precisely such ratios of components.

A sentence is added to the introduction part (line 101) explaining that different ratios of the CA and SA were tested to investigate if there are any dose-response effects of increasing ALA and cetoleic acid, respectively.

  1. Results for camelina oil and sandstone oil in various ratios should be compared with mixtures of other oils in more detail.

Different compositions of fatty acids in various oils can potentially lead to different effects in vivo. Including other oils than the ones used in our study (CA, SA) was therefore not relevant. Selection of a good control is also difficult, and this is the reason why a dose-response design using different ratios of CA and SA was chosen. In this way we wanted to investigate the effect of increasing levels of ALA and cetoleic acid, respectively. The control used is the standard rat diet.

  1. It is desirable to indicate for comparison what other factors and substances can influence the synthesis and deposition of EPA and DHA in rats.

Other factors that can influence the synthesis and deposition of EPA and DHA are already described in line 65-75 in the introduction and line 468 of the discussion.

  1. Please cite: 10.3390/molecules27186129.

This study “An Experimental and Theoretical Study on Essential Oil of Aethionema sancakense: Characterization, Molecular Properties and RDG Analysis” characterizes a plant oil high in the omega-6 fatty acid linoleic acid, and moderate in ALA (12%). Except for the content of ALA we have difficulties seeing how this paper is relevant to our study.  

  1. The article has a voluminous part "Discussion", but there is no "Conclusions". It is desirable to move some generalizations and main results to "Conclusions".

A section of the discussion is modified and transferred to a “Conclusion” paragraph.   

Reviewer 2 Report

The authors evaluated the effects of diets with different ratios of CA and SA on fa/fa rat metabolism.

- The conclusion in the abstract is too strong an not based on the actual data

- There is no clear hypothesis. What relevant health effects were expected?

- Power caclulation is missing

- I the intriduction the authors suggest that the conversion capacity of ALA to DHA is more efficient in females. So whre is the ligic in usin male rats only? 

- On what previous data did the author base the age of 10-12 weeks?

- On what previous data did the authors base the 4 weeks duration of the dietary intervention?

- The authors used parametric analyses - were the conditions of normal ducstribution evaluated and met? 

- The legends of the figures state n=6 but this is wrong for at least one group.

-line 344: not clear why erucic acid from CA is stressed since the % is 0.7 in both CA ad SA.   

-line 352 lowest dietary levels should be lowest ratios (or sentence should be re-written)  

- What % of citizins are currently on strictly plant-based diets in different eurpean countries?

- Line 404-406: it is not clear on what data obtained in thise experiments the authors base the second part of thier analyses (line 406: especially....).

-line 409-418: very strong causal inferences for a rather isolated and not very strong finding.  Additional data are needed. To allow this conclusion they could/should e.g. have included diets with CA supplemented with only (purified) cetoleic acid. 

- the association mentioned in lines 414-415 (dietary FA- blood FA) is missing in Table 3. 

- The assocations in Table 3 (simple pearson correlations) are not really informative because of the strong associations between e.g. 22:1 n-11,  20:5 n-3 and 22:6 n-3 in the diets (see table 2).  At least attampts should have been made to take this into account.

- lines 475-479: an odd combination of inforamtion becasue cetoleic acis is not detected in CA (table 2).  

Overall,  the purpose and hypothesis of this work does not really become clear and the experimental setup does not really allow for the inferences that are made. 

There are several sentences that are incomprehensible and likely grammatically incorrect. e.g. L391-393

The grammer when using e,g, adjectives is often incorrect ( -ly)

Author Response

  • The conclusion in the abstract is too strong an not based on the actual data

The conclusion of the abstract is modified.

  • There is no clear hypothesis. What relevant health effects were expected?

A hypothesis is added to line 94 in the introduction. The expected health effects are linked to increased synthesis and deposition of EPA and DHA.

Obese Zucker rats have a lower activity of FADS1 and FADS2 than a lean rat, which make it potentially easier to detect a stimulation of these enzymes involved in the synthesis of EPA and DHA. 

  • Power calculation is missing:

These oils are not previously tested in a study with Zucker rats. The number of animals per group are based on earlier studies of protein or oils like Wergedahl, H., Gudbrandsen, O. A., Røst, T. H., & Berge, R. K. (2009). Combination of fish oil and fish protein hydrolysate reduces the plasma cholesterol level with a concurrent increase in hepatic cholesterol level in high-fat–fed Wistar rats. Nutrition, 25(1), 98-104. The current study will provide a basis for power calculation for future studies.

  • I the introduction the authors suggest that the conversion capacity of ALA to DHA is more efficient in females. So where is the logic in using male rats only? 

It is true that it would have been interesting to include both genders, especially since a previous study has shown that females have a higher conversion capacity in humans. But the fact that males have lower capacity than females make it even more interesting to see if the capacity can be improved. The following sentences were added to the discussion (400-402):

Only male rats were included in the present study. The lower capacity in males compared to females previously observed in humans, might indicate a potential for improvement of EPA and DHA synthesis in males.

  • On what previous data did the author base the age of 10-12 weeks?

The activity of FADS1 and FADS2 is changing by age showing lower activity at the age of 6 weeks and more stable at the age of 9-12 weeks (Blond et al., 1989, https://pubmed.ncbi.nlm.nih.gov/2755316/)

  • On what previous data did the authors base the 4 weeks duration of the dietary intervention?

Duration of 4 weeks are used in many intervention studies and also by us (Wergedahl, H., Gudbrandsen, O. A., Røst, T. H., & Berge, R. K. (2009). Potent nutrients have shown good effect after 3-4 weeks in Zucker rats.

  • The authors used parametric analyses - were the conditions of normal distribution evaluated and met?

Normal distribution was tested by the Shapiro wiik test.

The data is analyzed by variance analysis, which is a robust test for this type of data.

  • The legends of the figures state n=6 but this is wrong for at least one group.

This is correct, the figure legends are corrected.

  • line 344: not clear why erucic acid from CA is stressed since the % is 0.7 in both CA and SA. 

The text is corrected to include both CA and SA.

  • line 352 lowest dietary levels should be lowest ratios (or sentence should be re-written)  

The text is corrected.

  • What % of citizins are currently on strictly plant-based diets in different eurpean countries?

The number of vegetarians (% of population) differs, and the definition of vegetarianism throughout Europe is not uniform. However, in 2022 most of the countries had 5-10% vegetarians included Belgium, Czech Republic, Estonia, France, Hungary, Ireland, Italy, Latvia, Lithuania, Netherlands, Norway, Poland, and Switzerland. Denmark, Finland, Germany, Sweden, and United Kingdom were on the top with 10-12 % vegetarians, while Greece, Portugal, Spain, and Slovenia were on the bottom with less than 5% vegetarians. In addition, a small number (1-4%) of the population in most of the countries are vegans. On the other hand, the numbers of people who eat or want to eat more plant-based food are increasing. In January 2022, Google revealed that searches for “vegan food near me” had dramatically increased in 2021 and attributed in to “breakthrough status”, meaning it increased by 5,000 percent or more indicating the rising popularity of vegan diets.

  • Line 404-406: it is not clear on what data obtained in thise experiments the authors base the second part of thier analyses (line 406: especially....).

The text (411-414) is modified to:

A positive effect of the almost four times higher dietary ALA levels in the CA than the Ctrl is shown by the 4.2-4.5 times higher EPA, 1.8-2.6 times higher DPA and 1.3-1.5 times higher DHA in the blood cells, liver and heart of the CA than the Ctrl. Compared to the Ctrl, the CA group showed significantly higher DPA (blood cells and heart) and DHA (blood cells) . Conversion of ALA to longer chain n-3 fatty acids have been confirmed in other studies of rats [24-28,53].

  • line 409-418: very strong causal inferences for a rather isolated and not very strong Additional data are needed. To allow this conclusion they could/should e.g. have included diets with CA supplemented with only (purified) cetoleic acid. 

We have previously done both in vitro cell studies with primary salmon hepatocytes and a human hepatocyte cell line, as well as an in vivo study with salmon, showing increase in EPA and DHA synthesis and deposition in salmon muscle with cetoleic acid (Østbye et al, 2019, Br.JNr). In the present study we wanted to investigate the dose effect of a combination of two dietary oils – one high in ALA and one high in cetoleic acid – on EPA and DHA levels in blood and liver/heart – and not the effect of pure cetoleic acid, which is very expensive to synthesise.

The following text is added to the paragraph (429-434):

Increasing in vivo content of EPA and DHA can partially be explained by the content of these fatty acids in the diets. In addition, the content of the bioactive cetoleic acid is increased by the inclusion level of SA in the diets, and can explain a higher levels of EPA, DPA and DHA in the rats. In vivo and in vitro studies have shown that cetoleic improves the EPA and DHA synthesis in salmon and human liver cells and increases deposition of these fatty acids in salmon muscle [40].

  • the association mentioned in lines 414-415 (dietary FA- blood FA) is missing in Table 3. 

The correlation value in the text is removed due to the low association level.

  • The assocations in Table 3 (simple pearson correlations) are not really informative because of the strong associations between e.g. 22:1 n-11,  20:5 n-3 and 22:6 n-3 in the diets (see table 2).  At least attampts should have been made to take this into account.

The text is modified to include the aspect of EPA and DHA content of the SA, and the correlation of these fatty acids with cetoleic acid in the oil.

  • lines 475-479: an odd combination of information because cetoleic acids is not detected in CA (table 2).  

Text is modified.

  • Overall, the purpose and hypothesis of this work does not really become clear and the experimental setup does not really allow for the inferences that are made. 

The hypothesis of the study is improved and changes made to the text.

  • Comments on the Quality of English Language, There are several sentences that are incomprehensible and likely grammatically incorrect. e.g. L391-393, The grammer when using e,g, adjectives is often incorrect ( -ly).

The quality of the English language is improved by the use of MPDIs editing service.

Round 2

Reviewer 2 Report

The authors have appropriately dealt with my remarks